# Exploring Stroke Patients’ Needs after Discharge from Rehabilitation Centres: Meta-Ethnography

**DOI:** 10.3390/bs12100404

**Published:** 2022-10-20

**Authors:** Basema Temehy, Sheeba Rosewilliam, George Alvey, Andrew Soundy

**Affiliations:** School of Sports, Exercise and Rehabilitation Sciences, University of Birmingham, Birmingham B15 2SA, UK

**Keywords:** stroke, cerebrovascular disease, needs, requirements, discharge, long term, qualitative

## Abstract

Healthcare providers must consider stroke survivors needs in order to enable a good quality of life after stroke. This review aimed to investigate the perceived needs of the stroke survivors across various domains of care following their discharge from hospital. A meta-ethnographic review of qualitative studies that reported needs of stroke patients after discharge from rehabilitation services was conducted. Main searches were conducted on the following electronic databases: Ovid Medline (1946 to 2021), CINAHL plus (EBSCO), AMED (EBSCO), PsycINFO (1967 to 2021), the Cochrane Library, and PubMed in June 2022. Main outcomes were related to stroke survivors’ views, experiences, and preferences on physical, psychological, social, rehabilitation needs, and other identified needs. Twenty-seven studies were included in the final analysis. The findings show that existing rehabilitation provision for stroke survivors does not address the long-term needs of stroke survivors. Two main issues were revealed concerning the unmet needs of stroke survivors: (1) a lack of information availability and suitability and (2) inadequacy of care and services. It is crucial to further investigate the needs of patients in Asian countries and the Middle East as there is very limited understanding of patients’ needs in the community in these regions.

## 1. Introduction

A stroke can occur suddenly but result in chronic effects in terms of a person’s physical health, emotional health, and social aspects [1]. Around 6.7 million people each year die of a stroke and a stroke is considered the second leading cause of deaths globally [2]. Strokes cause chronic disability; therefore, survivors and their families can experience consequences in the long term. For patients, these include cognitive disorders, concentration and memory difficulties, psychological issues, and severe physical disabilities [3]. The number of people having consequences following a stroke (ill health, disability, deaths) is likely to increase twofold before 2030 [4].

Around 80 per cent of those who survive a stroke are sent home from the hospital to continue to recover [5]. Following discharge, an increasing number of people who have had a stroke live with disability; in 2013, there were 113 million disability-adjusted life-years due to stroke globally [2]. Those who suffer a stroke face a difficult process of recovery, whereby they need to improve their functioning, including speech, physical, and cognitive functioning [6]. Yet, following discharge, stroke survivors have various needs that often go unmet and they feel neglected [7]. The definition of unmet needs post rehabilitation is where patients have a need for something that would help them to recover from a stroke but that which is not being provided [8,9]. For example, current research [1] has identified that only three out of ten people with strokes are receiving the recommended number of review assessments (at 6 weeks, 6 months, and 12 months) following discharge.

Practical evaluation of unmet needs can be undertaken by determining whether any help has been provided for particular needs identified by people with strokes and whether this issue has been dealt with in a sufficient manner [9]. Qualitative research is well-positioned to detail such needs, and there is a large scale of qualitative studies which have explored the needs of people with strokes post-discharge. Studies which address stroke needs vary in the period explored, from one month [10] to ten years post-stroke [11]. The needs addressed are across various domains of care, such as rehabilitation needs [12], psychological and emotional needs [13], or perceived information needs [11,14].

A broader understanding of stroke survivors’ needs has not been reviewed after they are discharged from healthcare services from the perspective of both patient and professional. Existing systematic reviews have focused mainly on specific types of needs that are perceived by both stroke survivors and carers. Examples of perceived needs include educational needs and rehabilitation needs [15,16,17]. Other reviews have been limited by their inclusion of particular design types or focusing on one specific group of participants. For instance, past reviews have included survey studies involving just stroke patients [6] or have included quantitative studies only of community-dwelling stroke survivors [18]. Existing syntheses [19] of qualitative evidence have not corroborated the needs of stroke survivors with the professionals’ perspectives, and this review focused purely on participants from community settings only. Review-based research is needed that can triangulate the experience of different patients and healthcare professionals. This is important because the chronic nature of stroke impacts requires the collaborative input of the patient and the health professional to identify unmet needs and to work together for optimal recovery. Meta-ethnography provides a conceptual framework that goes beyond the simple aggregation of primary findings and is a particularly suitable method to provide new insights into the unmet needs of stroke survivors. To the best knowledge of the researchers of the present paper, no meta-ethnography study has summarised the literature in this area. Therefore, this review aimed to explore the needs of stroke survivors post-discharge from the hospital in aspects across various domains of care from the perspectives of the patients and professionals.

### Review Question

Following their discharge from the hospital, what are the needs of the stroke survivors across various domains of care perceived by patients and healthcare professionals?

## 2. Methodology

### 2.1. Study Design

This review was previously registered (PROSPERO 2021 CRD42021256405).

Meta-ethnographic guidelines have supported the methodological reporting of this review. This includes the original guidance from Noblit and Hare [20] as well as considerations from recent guidelines [21,22]. Meta-ethnography is a systematic comparison of conceptual data found in primary qualitative research in order to establish and develop ideas, concepts, theories, and models. Cahill et al. [23] explain that a new generation of concepts is needed to explain the relationships between findings as opposed to just describing the data. To perform this, the researcher can keep a diary of questions to answer. This type of research was intended to retain the meanings and contexts in the original studies, yet create conceptual models and theories in the realm of study [20].

### 2.2. Search Strategy

The literature review search comprised three parts. The first part was to clarify the focus area of the study. To perform so, a scoping search was undertaken. Following this, the primary searches were carried out in June 2022 on the following electronic databases: Ovid Medline (1946 to 2021), CINAHL plus (EBSCO), AMED (EBSCO), PsycINFO (1967 to 2021), the Cochrane Library, and PubMed. The keywords used within the databases were identified using the PICOS strategy, with alternative spellings and synonyms also searched for (see Appendix A). Boolean terms ‘OR’ and ‘AND’ were utilised, along with subject headings (for example, MeSH). A librarian supported the process, and this was checked. Following retrieval of the papers, reference lists and bibliographies were searched manually to find any further studies of interest. The grey literature of PhD dissertations and conference papers were considered. Main search terms were searched for in Google Scholar and the Science Direct website, and the first 20 pages of the results were screened. For the included studies, the researchers sought the profiles of study authors on ResearchGate and Academia.edu. Appendix A contains an example of the search carried out in Medline.

### 2.3. Inclusion and Exclusion Criteria

Eligibility criteria are presented according to the SPIDER (Sample, Phenomena of Interest, Design, Evaluation, and Research Type) acronym.

#### 2.3.1. Sample

Studies were included if they represented people with strokes. The target population for the study was adults who had been discharged from hospitals after a stroke, ischaemic stroke, or haemorrhagic stroke, who had finished rehabilitation (either at a rehabilitation service or in the community). Studies that involved a mixed sample, such as patients with traumatic brain injury and stroke, were included if the data for stroke survivors could be extracted. Studies were also included that considered post-discharge needs of people with strokes from the perspective of caregivers and health professionals.

#### 2.3.2. Phenomena of Interest

Studies were included if they were able to consider the needs of stroke survivors after they had been discharged from rehabilitation. Studies could focus on exploring various needs, for instance physical, psychological, social, political, cultural, environmental, or rehabilitation. Needs had to be reported by the stroke patients themselves and during the post-discharge, subacute, or chronic phases. Mixed method studies were included if the qualitative data could be extracted. Studies that were excluded were those where stroke patients had their needs assessed prior to discharge, those that assessed patient satisfaction rather than perceived needs, those studies which solely assessed carers’ needs, and mixed studies where the qualitative data could not be extracted.

#### 2.3.3. Design

All types of qualitative methodology were considered; for instance, this included types of grounded theory, types of phenomenology, types of narrative research, and descriptive or interpretive designs. A mixed methods study that included a clearly identifiable qualitative phase and reported that phase in a way that represented a form of qualitative design were included. Case studies were excluded.

#### 2.3.4. Evaluation

All types of methods used in qualitative studies were acceptable; for instance, this could include interviews, observations, field diaries, vignettes, or surveys with open questions. Studies that quantised data or restricted reporting of experiences were excluded.

#### 2.3.5. Research Type

Qualitative or mixed methods designs were included.

#### 2.3.6. Other

Studies that were conducted in hospitals, nursing homes, or the community.

There was no limitation on dates of publishing. Only articles in English were considered and studies had to be related to humans and have an abstract.

### 2.4. Study Selection

Two independent reviewers carried out the study selection process. The titles and abstracts were identified following a search and screening process, where any duplicates were removed. The eligible papers were saved on EndNote. Following the creation of the abstract shortlist, the two independent reviewers (BT and GA) screened the articles using the inclusion and exclusion criteria and compared their results. Disagreements were resolved by a third reviewer. The full text of the papers were read to shortlist the relevant articles.

### 2.5. Quality Appraisal and Certainty Assessment

Quality appraisal was undertaken using the Joanna Briggs Institute qualitative critical appraisal checklist. The certainty assessment was carried out by using the GRADE-CERQual assessment tool. The quality of each paper was assessed using two independent reviewers (BT and GA) and a third reviewer (SR) in the case of any disagreements.

### 2.6. Data Extraction

A pilot data extraction was conducted on two papers [24]. Following this, the data were extracted by two independent reviewers, and any discrepancies were ratified by a third reviewer. The following data were extracted from the papers studied: author, publication date, country of study, aim, features of the sample (such as sample size, age, gender, and length of time since discharge), study setting, design, and results. Similarly extracted was information regarding patient needs in terms of the physical, psychological, social, rehabilitation, financial, and other aspects. Data were documented in Microsoft Word tables. Authors of studies were contacted in the case of missing or unclear information in the included papers.

### 2.7. Data Analysis

An independent reviewer undertook the meta-ethnographic synthesis according to recommended guidelines for synthesising the data [20,21]. Studies were read multiple times in chronological order by the researchers, who determined the relationship between the studies by addressing the studies’ design, aims, setting, and sample characteristics, for example, age, socioeconomic status, gender, ethnicity, and the length of time since stroke event. Following this, new concepts and metaphors were identified. In this stage, line by line coding of all findings of primary studies was the technique that was used. Then, theses codes were juxtaposed and brought together to create clusters and themes by grouping the interpretation of first order construct (participants’ views and interpretations reported in the included studies) and second order constructs (authors’ interpretations of participants’ views in these studies). The relationships between second order constructs were then used to create the third order constructs. These associations between the studies occurred in two ways—reciprocal, for overlapping studies, and refutational, for conflicting studies. Tables and grids have been utilised for this purpose. A senior reviewer reviewed the findings of data synthesis (BT). An audit trail can be seen in Appendix A.

## 3. Results

In total, 5345 records were identified from the database search. Forty-three further records were identified from searching references. Following the removal of duplicates, there were 4627 articles, of which, 127 articles were assessed for eligibility. Ninety were excluded and thirty-one were included in the critical appraisal stage (Appendix A). Four studies were excluded due to low quality. Twenty-seven articles were included in the final analysis. The results of the search and screening process are presented in the PRISMA diagram (Figure 1).

### 3.1. Study Characteristics

Of the twenty-seven included studies, eleven were conducted in the UK, six in Canada, five in Australia, two in the US, and one in each of China, Iran, Malaysia, and Norway. The sample size of included studies ranged from six participants [25] to one hundred twenty-five participants [13]. Among these studies, thirteen articles [10,11,26,27,28,29,30,31,32,33,34,35,36] recruited stroke participants only. Mixed participants of stroke survivors and carers were recruited in nine studies [14,25,37,38,39,40,41,42,43]. Three studies [12,13,44] included health professionals as well as stroke survivors and carers. Nordin et al. [45] and Abrahamson and Wilson [46] recruited health professionals and patients. The characteristics of studies and patients can be found in Table 1.

### 3.2. Quality Assessment and GRADE

The quality assessment is included within the GRADE-CERQual assessment following the synthesis section (Table 2). A breakdown of the quality assessment can be seen in the Appendix A.

**Table 1 behavsci-12-00404-t001:** Characteristics of studies.

**Authors**	**Geographical Location**	**Aim**	**Methodology**	**Participant**	**Time Since Stroke**
[10]	China	To identify the rehabilitation needs of Chinese elderly patients following a stroke	Qualitative ethnographic approach. Semi-structured interview.	Fifteen stroke survivors Nine females Six males	One week before discharge from the rehabilitation ward and one month after discharge.
[11]	Sweden	To use patient journey mapping to explore post-discharge stroke patients’ information needs to propose eHealth services that meet their needs throughout their care and rehabilitation processes.	Qualitative research. Focus groups.	Young (<65 years) and old (≥65 years) stroke patients Female: seven Male: five	One focus group included patients with strokes more than 10 years ago. Two groups included patients with strokes less than 10 years
[12]	Canada	The aim of this study was to examine the rehabilitation needs of this clientele from their hospitalisation to their reintegration into the community.	Qualitative research tool was selected. The method of focus group discussion.	The patients (n = 4) Caregivers (n = 5) Healthcare providers (n = 9) Administrators (n = 7) Gender: not provided	Three patients with strokes from 2 to 3 years. One patient had a stroke from 4 to 8 years.
[13]	UK	To explore patients’, carers’, and health professionals’ experiences of psychological need, assessment, and support post-stroke while in hospital and immediately post-discharge.	Exploratory study. Qualitative semi-structured interviews and focus groups.	Thirty-one stroke patients, twenty-eight carers, and sixty-six health professionals. Male: eighteen patients, nine carers Female: thirteen patients nineteen carers Health professionals’ genders were not provided.	Mean length of 171.23 days between discharge and interview.
[14]	Australia	To identify patients’ and carers’ perceived barriers to accessing and understanding information about strokes.	Semi-structured interviews at two points in time.	Initial interviews were conducted with 34 stroke patients and 18 carers, and follow-up interviews were completed with 27 patients and 16 carers. Fourteen female patients Thirteen female carers	Prior to and 3 months following discharge from an acute stroke unit.
[25]	Canada	To explore the stroke education perspectives in a Canadian rehabilitation centre to illustrate one approach for addressing this problem.	Qualitative description study was overlaid by phenomenology. Face-to-face semi-structured interview.	Three patients and three caregivers. Three male patients Three female caregivers	Not identified.
[26]	UK	To explore stroke survivors’ needs and their perceptions of whether a community stroke scheme met these needs.	A qualitative study using a phenomenological approach. Semi-structured face-to-face interviews.	Twelve stroke survivors. Female: five Male: seven	Mean of 26 months post-stroke.
[27]	Australia	To explore the experiences of community-dwelling stroke survivors at one, three, and five years using a community-based, cross-sectional study.	A modified, grounded theory approach. Semi-structured interview.	Ninety-one stroke survivors at one, three, and five years after stroke. Forty-seven males Forty-four females	Cohort One: People who had had a stroke 1 year prior to recruitment. Cohort Three: People who had had a stroke 3 years prior to recruitment. Cohort Five: People who had had a stroke 5 years prior to recruitment.
[28]	Iran	To illuminate how stroke survivors experience and perceive life after strokes.	Grounded theory approach using semi-structured interviews.	Ten stroke survivors. Male: six Female: four	Patients had strokes within the past 3–6 months.
[29]	UK	To investigate how contextual factors, as described by the World Health Organisation’s International Classification of Functioning, Disability, and Health (ICF), impact stroke survivors’ functioning and how needs are perceived in the long term after strokes.	Semi-structured, in-depth interviews.	Thirty-five stroke survivors. Males: 49% Females: 51%	Total of 49% had strokes within 1 to 2 years. Total of 31% had strokes within 3 to 5 years. Total of 6% had strokes within 6 to 8 years. Total of 14% had stroke more than 9 years.
[30]	UK	To investigate how younger stroke survivors’ experiences of care are shaped by the field of stroke and how, in navigating stroke care, individuals seek to draw on different forms of capital in adjusting to life after strokes.	One-to-one, semi-structured interviews.	Thirty-one stroke survivors were interviewed. In ten interviews, carers also took part. Nineteen males Twelve females	Patient had stroke within 6 weeks and 28 months.
[31]	Norway	To explore young and midlife stroke survivors’ experiences with the health services and to identify long-term follow-up needs.	This qualitative study applied a hermeneutic phenomenological approach. Two cohort, in-depth interviews.	Sixteen stroke survivors. Five females Eleven males	Patient had stroke within 1.5 to 10 years after stroke onset.
[32]	Sweden	To explore stroke survivors’ experiences of healthcare-related facilitators and barriers concerning return to work after stroke.	Qualitative study. Focus groups.	Twenty stroke survivors. Seven females Thirteen males	Patient had been referred to stroke rehabilitation within 180 days after stroke onset.
[33]	Australia	To examine the unmet needs of younger stroke survivors in inpatient and outpatient healthcare settings and identify opportunities for improved service delivery.	Qualitative descriptive approach. In-depth, semi-structured interviews.	Nineteen young stroke survivors. Ten females and nine males.	Patients had stroke within 6 months to 24 years.
[34]	Canada	To identify the educational needs of older adults who have had a stroke in order to support their participation in leisure activities that promote cognitive health.	A descriptive study. Mixed-methods design was used with an emphasis on qualitative data and involving semi-structured interviews.	Twenty people. Fourteen males Six females	Mean of 8 months post-stroke. Mean of 5.9 months post-discharge.
[35]	Australia	To explore the needs and experiences of people who cease driving following a stroke with the aim of informing clinical practice.	Qualitative phenomenological approach. Semi-structured interviews.	Twenty-four stroke participants. Seventeen males Seven females	Mean of 5 years post-stroke.
[36]	UK	To develop local stroke services by involving, in a meaningful way, those affected by stroke in identifying and prioritising service development issues.	An action research framework. A combination of semi-structured interviews and focus groups with both patients and carers.	N=35 Patients recruited from hospitals (n = 30) Females: 53% Patients recruited from community (n = 5) Females: 52%	Not identified.
[37]	UK	To identify the information needs of patients and their informal carers at various stages post-stroke with the aim of developing a database from which individualised information packages could be provided.	Grounded theory approach. In-depth, qualitative, semi-structured interviews.	Nine were interviews with patients, ten were interviews with patients and carers together, and two were interviews with carers only (totalling thirty-one people in all). Eleven were male and ten were female.	Seven interviews were carried out with patients and/or carers immediately post-stroke. Five immediately post-discharge. Nine between 2 months and 1 year post-discharge.
[38]	UK	To identify the long-term support needs of patients with prevalent stroke, and their carers identified from practice stroke registers.	Qualitative study. Focus groups.	Twenty-seven patients and six carers subsequently participated in the focus groups/interviews. Nineteen females Fourteen males	Median of 4 years post first stroke. Median of 2.5 years since last stroke.
[39]	US	To study the perspectives and experiences of stroke survivors and partners of stroke survivors regarding sexual issues and perceived rehabilitation needs.	Qualitative, exploratory, Individual, semi-structured interviews	Fifteen stroke survivors and fourteen partners of stroke survivors. Sixteen males Thirteen females	Patients: median of 45 months post-stroke. Partners: median of 51.5 months post-stroke.
[40]	USA	To examine rural Appalachian Kentucky stroke survivors’ and caregivers’ experiences of receiving education from healthcare providers with the long-term goal of optimizing educational interactions and interventions for an underserved population.	Qualitative descriptive study. Semi-structured interviews.	Thirteen stroke survivors and twelve caregivers. Sixteen females Nine males	Mean of 3.6 years post-stroke.
[41]	Canada	To explore the experiences and needs of Chinese stroke survivors and family caregivers as they return to community living using the Timing it Right Framework as a conceptual guide.	Qualitative interviews In person or telephone interviews depending on the participant’s preference.	Eighteen participants including five stroke survivors and thirteen caregivers. Nine females Nine males	Patients: median of 6 months post-discharge. Caregivers: median of 8 months post-discharge.
[42]	Canada	To report the experiences and perceptions of people with stroke and their caregivers in the existing continuum of stroke care, social services, and rehabilitation in the province of Québec (Canada).	Phenomenological qualitative study. Focus groups.	Sixty-eight participants were recruited and attended the ten focus groups. Thirty-seven stroke patients and thirty-one carers. Twenty-nine males Thirty-nine females	Mean of 2.6 years post-discharge.
[43]	Australia	To explore community-dwelling first-time stroke survivors and family caregivers’ perceptions of being engaged in stroke rehabilitation.	An interpretive study design. Face-to-face using a semi-structured interview.	Twelve and ten caregivers. Twelve males Ten females	Not identified.
[44]	Canada	To gain insight into healthcare and social structures from the perspective of patients and caregivers that can better support long-term stroke recovery.	Qualitative descriptive design. Semi-structured interview.	A total of twenty-four participants were recruited: sixteen stroke survivors (female = five, male = eleven, aged 48–87), four spouses, (females aged 62–80), three stroke recovery group coordinators (female), and one speech pathologist (female).	Mean of 8.74 years since stroke.
[45]	Malaysia	To explore the perception of rehabilitation professionals and people with stroke towards long-term stroke rehabilitation services and potential approaches to enable provision of these services.	Qualitative study using focus groups.	Fifteen rehabilitation professionals. Eight stroke survivors. Fourteen females Nine males	Patients had stroke from 1 to 2 years.
[46]	UK	This study explored needs identified by patients, how they were addressed by the six-month review (6MR), and whether or not policy aspirations for the review were substantiated by the data.	Philosophy: critical realism. Design: multiple case study design. Methods: interviews.	Forty-six patients and twenty-eight professionals. Gender was not provided.	Patients and carers were interviewed at about six weeks post-discharge after their 6MR and, where possible, after their annual review.
**Authors**	**Findings**				
[10]	Five themes: informational needs; psychological needs; physical needs; social needs; spiritual needs.				
[11]	Five themes: A holistic view of the care process; understanding the illness; collaboration with care providers; tracking the rehabilitation process; practical guidance through healthcare and community services.				
[12]	Nutrition; body condition; personal care; communication; housing; mobility; responsibilities; interpersonal relationships including sexuality; community living; leisure activities; psychological; cognitive.				
[13]	Two themes: Minding the gap and psychological expertise.				
[14]	Three themes: limited availability and suitability of information; the hospital environment; patient and carer factors.				
[25]	Five themes: secondary prevention; rate of recovery; knowledge collection; transition to home; adherence to home programme.				
[26]	Three themes: creating a social self; provision of ‘responsive services’ in the community ; informal support network.				
[27]	Three themes: knowledge about stroke; communication with the health system; influences on transition home.				
[28]	Two themes: functional disturbance and lack of social support.				
[29]	Environmental factors; support and relationships; products and technology; services, systems, and policies; attitude; personal factors; life experiences; social position; personal attitude.				
[30]	Four themes: healthcare professional as expert; expectation of involvement in care; social capital; variations in economic capital.				
[31]	Two themes: difficulties accessing health services and lack of tailored follow-up services.				
[32]	Two themes: requesting rehabilitation planning, healthcare information, and coordination and increased support in daily life would facilitate return to work.				
[33]	Three themes: inadequately addressed psycho-emotional and cognitive needs after young stroke; isolation from lack of information and structured support; failure to deliver age-relevant patient-centred care.				
[34]	Three themes: activities perceived to be beneficial in promoting cognitive health; continuity versus changes in participation post-stroke; factors influencing leisure participation.				
[35]	Four themes: life without driving; key times of need; alternatives and other ways; carer support and assistance.				
[36]	Four themes: prevention; immediate care; early and continuing rehabilitation; transfer of care and long-term support.				
[37]	Three themes: clinical information; practical information; information on continuing care and resources in the community.				
[38]	Three themes: psychological and emotional problems; information needs; contact with services.				
[39]	Seven themes: sense of loss and functional changes affect sexuality; relationship changes affect sexual functioning; difficult to talk about sex; little or no discussion of post-stroke sexuality by rehabilitation professionals; need to tailor education about sex to the individual/ couple; timing is key in presenting information about sex after stroke; provider rapport and competence is vital to discussing sexual issues.				
[40]	Five themes: providers of education; receivers of education; content of education; delivery of education; timing of education.				
[41]	Two themes: information and training needs of stroke survivors and caregivers change over time, and Chinese resources are needed across care environments.				
[42]	Four themes: accessibility of care; appropriateness of care; expertise of the healthcare workers and continuity of care				
[43]	These themes: readiness to return home; coping with care transition; dealing with fragmented rehabilitation services and uncertainty about ongoing rehabilitation.				
[44]	Two themes: experiences of managing stroke and resources for support.				
[45]	Four themes: the needs for continuity of care; beliefs about long-term rehabilitation; perceived barriers to long-term stroke rehabilitation; approaches to long-term rehabilitation.				
[46]	Two themes: perceived needs for community stroke rehabilitation and perceived need for information, education, and support.				

Two major themes related to the needs of stroke survivors emerged from the data. These included: (1) limited availability and suitability of information, (2) Adequacy of care and services.

#### 3.2.1. Major Theme One: Limited Availability and Suitability of Information

This theme contained three subthemes including:

Subtheme One: information needs related to stroke pathology.

This includes stroke definition, symptoms, signs, causes, treatment, complication, and stroke recurrence.

The finding in this review revealed a lack of pre stroke information of stroke-risk factors and warning signs [27,36,38,41]. Poor pre stroke information leads to failure of participants to respond to their stroke symptoms [27]. Yueng et al. [41] described educational resources can help stroke survivors and families learn about stroke including TV, radio, flyers, and newspapers.


*I cannot understand what caused it to happen … I did not know what a stroke was.*
[27] (p. 85)

After stroke, detailed information on prevention of stroke recurrence was desired [32,36,37,46].


*That [information] was fairly zero, actually! I would have liked more information about how to prevent another stroke and also … any alarm signals.*
[46] (p. 5)

Subtheme Two: information needs regarding stroke recovery.

Participants needs information on stroke treatment including information related to emergency intervention, medication, medication side effects, treatment plan, recovery rate, recovery facilitators, and guidance on available health services.

Participants in the included studies explained that, during the immediate post-stroke phase, they needed a great deal of information regarding the subsequent course of action and details of those who can be contacted for help [36,40].


*“I would like to know what services were available, you know.”*
[27] (p. 87)

Subtheme Three: the means of information delivery, including, the amount, relevance, time, format, and language of information.

##### Information Delivery Methods

Amount: One study revealed that healthcare professionals provided insufficient amount of information to stroke patients. Although some information is provided, it was often vague or lacked information specific to the nature of their health status [14].

Relevance: In terms of information content, participants indicated that the standard information packs that they were given had limited relevance, with some patients being given information about acquired brain injuries or all-age stroke groups. Most participants reported a need to receive information that was more relevant to their age group and health diagnosis [14,26,33].


*“didn’t have relevant brochures … not a lot of detail.”*
[14] (p. 72)

Timing: Participants expressed needs for receiving information multiple times across the continuum as information can be difficult to absorb immediately post-stroke due to the stroke side effects, feeling overwhelmed, a chaotic environment, rehabilitation commitments, and memory deficiencies [14,35,46].


*“Participants described the need for multiple repetitions of education over time, across the continuum of care, and into the chronic phase of stroke.”*
[40] (p. 20)

Format: There was a need for different types of education resources, such as written information (pamphlets, brochures, and binders), verbal information using group discussion and guest speakers, and visual teaching using videos and technology. Verbal delivery was the favoured mode for information delivery as it was suggested as easy to remember [40]. The information needs to be simple and presented clearly without using medical terms [14]. Further, stroke survivors expressed their needs to have access to their health records; patients wanted access to information post-discharge including the diagnosis, treatments, medications, lab results, referrals, appointments, and home visits [11].


*‘I need to learn, sometimes on radio and television they have programs about stroke recovery. I listen and use the information. Some guidelines are very important and can help us to improve our life style after stroke.’*
[28] (p. 251)

##### Challenges for Information Delivery

After accounting for the need to tailor information, further factors that influenced stroke patients’ experience of care were reported. These factors can be divided into patient factors and staff factors. Participants preferred healthcare providers who were proactive in their initiation of education. Participants did not seek information because they do not know what to ask, forgot to ask, felt they did not need information, or assumed that information would be given to them [14,40,46]. Poor health and stroke-related impairment also led patients not to seek information [14]. A lack of knowledge on behalf of staff and staff time constraints were the most reported barriers to deliver information [11,13,14,43].


*“You were lucky to get a doctor to come and speak with [you].”*
[14] (p. 74)


*“The doctors always said … please feel free to ring up … But it’s like all these things, you don’t know the questions to ask. You’ve no idea.”*
[46] (p. 5)


*“The physicians here at the hospital did not agree with the stroke rehab physicians about when I was going to start working.”*
[32] (p. 745)

#### 3.2.2. Major Theme Two: Adequacy of Care and Services

Rehabilitation process experience: In order to return to motor activity as well as a better quality of life, rehabilitation is critical. Several patients indicated that they felt anxious and distressed about the therapy processes that they went through during rehabilitation. In general, they seemed to perceive that rehabilitation services were fragmented and disorganised [11,33,43].


*“They don’t really help you get back into life, do they? They just sort of, you have a stroke, you have physio and that’s it.”*
[30] (p. 1915)

Transition experience at discharge: It was found in the papers that the transition in care from hospital to home was poor, disorganized, and fragmented [25,27,43]. Patients worried about coping, receiving support, and about whether they were ready to be discharged from the hospital [36]. To communicate their needs regarding education, patients needed to conduct a trial stay at home. This enabled them to resolve any issues, including with equipment [25]. In the study by Chen et al. [43], it was revealed that patients were exposed to a potential lack or breakdown of care as a result of ineffective handover among clinical providers/staff [43]. Patients feared that interrupted rehabilitation would thwart their recovery and stop them from becoming rehabilitated [43].


*“What they did was they sent me home a couple of weeks before time and they figured that … I would realise, you know, exactly what I would need so and that’s why they did it.”*
[25] (p. 283)

Participants had coping issues when transferred back into the community including physical restriction, cognitive issues, and feelings of abandonment and being a burden on others [10,28,42,43]. Participants were frequently unsatisfied due to various unmet needs.

Subtheme One: Intervention needs

It was reported that rehabilitation programmes need to be more frequent, intensive, long-term, and flexible to meet participants’ goals and be appropriate for their age and health situation. Participants wanted return to work preparation, driving assessment, individualised sex intervention, and resources to support social participation.


*“there’s no way [the] one hour a week that the government gives you is going to fix people.”*
[33] (p. 1701)

Further, there was a reported shortage in cognitive and psychological support. A lack of psychological support was reported in many of the papers [10,12,13,28,30,31,32,33,42,43]. Participants expressed the need for early and formal cognitive support, with rehabilitation not solely being focused on physical performance. In Harrison et al.’s [13] study, participants commented that a lack of formal psychological support early on could influence how much patients are rehabilitated and consequently recover.


*“it should have been a formal process to gain access to a psychologist” and it would have been beneficial from an earlier time point, such as from the acute hospital.”*
[33] (p. 1700)

Subtheme Two: Social needs

Most stroke survivors indicated that they had unmet social needs. Specific social needs after discharge included motivation, income support, and travel support.

1—Motivation: Several participants asserted that they looked for hope in their situations. They also highlighted the need for more positive discourse from healthcare professionals to help them recover as much as possible [28,31,33,35,44]. Many participants explained that, as their stroke became chronic, their motivation to carry out their previously learned exercises at home decreased [45].


*“Initially, I was motivated. After several months, I don’t feel that excited anymore.”*
[45] (p. 6)

1:1—Family support: A key part was played by participants’ families, who participated in caring roles, rehabilitation, and advocacy for improved services [26]. Carer training was one of the most reported needs [37,43,45]. However, difficulty identifying the main carer and being busy with their lives were barriers to caregivers’ training [45].


*“it was almost like [being] throwing in the deep end. I’ve never showered anyone in my life beforep … we just sort of muddled through … maybe like when you’re leaving rehab inpatient, someone probably should go through with the partner of the person about showering, medications.”*
[43] (p. 78)

1:2—Peer support: Stroke survivors stated that stroke recovery groups were an important factor for stroke recovery. Peer groups helped with psychological and emotional factors by having someone to talk to, sharing, understanding, learning from each other, and setting goals [33,44]. In spite of the advantages offered to patients by stroke recovery groups, survivors found that they had trouble accessing such groups after they had been discharged [44]. There are specific needs regarding stroke peer groups including age relevance and peers with similar conditions and interests.


*“I wish that there would be a therapy group for people who are in the same situation… how we can help each other, what kind of demands can I make at work. I would be open to that after 4 months, when I have come to terms with my situation a bit and it would’ve been OK to share it with others.”*
[32] (p. 744)

2—Income support: Many stroke patients lacked stability as a result of poor support from insurance or a lack of financial help [28]. The reported financial needs were divided into two. In terms of health insurance, stroke rehabilitation costs are high and not all patients were able to pay for the stroke rehabilitation aspects that they felt were required [44]. In addition, stroke survivors had financial needs in terms of mobility aids and house adaptations (wheelchairs, scooters, walking sticks, and frames).


*“Definitely, one of the most important necessities for every human being in the world is social insurance that supports people with stroke so they have a stable community.”*
[28] (p. 251)

3—Travel support: Another important issue highlighted by participants was a lack of transport; survivors lacked access to public transport to different locations or experienced long waiting times for public transport and often the seats were full [35]. Alternative transport options were found to be inconvenient due to high costs. Of those with their own transport, there were few parking spots at hospitals and the disabled spaces were often full [35,45].


*“often the seats on trains and buses were full, disabled parking spots were often taken and that there was increased waiting time associated with public transport.”*
[35] (p. 277)

Subtheme three: Planning

Patients reported needing a clear treatment plan. These plans need to be individualised and relevant to their age and personal history [26]. Further, participants want to be involved in planning and setting goals.


*“It would have been good to get a plan earlier. To receive [a rehabilitation] contact earlier.”*
[32] (p. 744)

Subtheme Four: Care continuity

1—Ongoing care: One of the most reported needs was continuity of care—participants expected care to be continued post-discharge. However, long waits for rehabilitation to commence post-discharge were reported in a number of the studies [11,26,28,29,30,31,33,38,42,43,45,46]. A primary barrier to continued care was seen as the inadequacy or lack of equipment/preparedness in rehabilitation wards and community rehabilitation [45].


*“It’s a long time to wait before they came round, I wanted to get moving because the physio was so good in hospital … but then when you come home there’s nothing … I wanted to just get going and build on what I was doing in the hospital.”*
[46] (p. 3)

2—Follow-up: Post-discharge follow-up was one of the most reported needs. Patients needed follow-up services to have feedback on their progress, to be closely assessed, and receive guidance on performing activities appropriately and safely [31,34]. Although some participants received follow-up services, the quality of the follow-up was not as expected; some regular meetings did not occur, there was a delay in replying, questions were not being addressed due to lack of expertise, and the follow-up was not tailored to their specific needs [31].

3—Communication: Stroke survivors have difficulty in contacting healthcare professionals to obtain support. Participants felt it would benefit their rehabilitation outside of the hospital setting if they had a coordinator they could discuss services and concerns with [43]. In addition, there was a need for trained healthcare professionals to deliver their diagnoses in a sensitive manner, as well as perform psychological assessments, sex assessments, and active listening, and respect the patients and their word choices. There was a lack of congruity between different healthcare actors [27,32].


*“You should get some kind of contact person… someone who calls and checks on you “how is it now, are you experiencing any problems?” could refer you to, well, here or there.”*
[32] (p. 744)


*“No automatic follow up on what I am doing. That’s the biggest problem. You’ve got to have some input…There is no follow up unless you do it yourself.”*
[43] (p. 80)

##### Needs of Younger Patients

Regarding younger participants’ rehabilitation needs, three studies [30,31,33] recruited younger participants (mean ages of 46, 48, and 41). In these three studies, participants express needs for psychological and emotional support, care continuity, return to work, and follow-up. Younger stroke survivors found rehabilitation care irrelevant to their individual goals and to be generic, fragmented, and designed for old patients [33]. Those participating in the study indicated the requirement for rehabilitation services that were age-relevant. In addition, some participants were not able to receive outpatient rehabilitation because their deficits were mild, or the majority of participants had to wait three to six months for an outpatient clinic appointment [33].


*“they just treated me like I was 70 years old but I’m actually 35” and “it’s like they put you into a box; you had a stroke, so this is how we deal with people that had a stroke.”*
[33] (p. 1701)

##### Carer Needs

Three studies [12,25,42] separated the needs for patients and carers. Although strokes have an impact on caregivers’ personal lives, they had difficulty in expressing their needs [42]. Most carer needs were equivalent to patients’ needs. However, training courses and psychological support were needs repeated by carers.

**Table 2 behavsci-12-00404-t002:** CERQual assessment.

Review Finding				CERQual Confidence
Limited availability and suitability of information	Needs for information on stroke pathology	Pre stroke information After stroke information		High confidence
	Needs for information on stroke recovery	Need for information on stroke rehabilitation		High confidence
		Need for guidance on health services		Moderate confidence
	Adequate information delivery	Amount		Moderate confidence
		Relevance		Low confidence
		Time		Moderate confidence
		Format		High confidence
Adequate care and services	Intervention needs	Programme Intensity		Low confidence
		Return to work		Low confidence
		Driving rehabilitation		Low confidence
		Sex rehabilitation		Low confidence
		Leisure activities		Low confidence
		Psychological support		High confidence
	Social needs	Motivation	Peer support	Moderate confidence
			Family support	High confidence
		Income support		High confidence
		Travel support		High confidence
	Engagement in goal setting	Clear plan		Low confidence
		Individualised plan		Low confidence
	Care continuity	Ongoing care		High confidence
		Follow-up services		High confidence
		Communication	Contact with health services	High confidence
			Coordinator	Low confidence
			Trained professionals	Moderate confidence

## 4. Discussion

In order to enhance post-stroke care, many researchers have turned their attention towards understanding the needs of stroke survivors and caregivers. Thus, the key objective of this review was to investigate the unmet needs of stroke survivors after discharge from the hospital. Throughout this review, two key themes emerged: (1) limited availability and suitability of information; (2) adequacy of care and services.

The findings of the present review indicate that stroke survivors have a number of diverse educational needs, and many of these needs are currently being unmet. The findings are very much in line with those revealed in Hafsteinsdóttir et al.’s [15] study. Patients who have suffered a stroke must be educated carefully because strokes can cause ischaemic injuries that have a significant impact on learning. One potential solution to meet the information needs of stroke survivors could be to assign a nurse as a personal educator. However, the nurse must be suitably trained in understanding risk factors, pathophysiology, the impacts of a stroke, the availability of community resources, and the emotional needs of patients and their families [47]. Participants in this study pointed out that professional expertise was required to manage their situations [27,31]. Thus, if a nurse does not have thorough and comprehensive knowledge of strokes, they may be unable to give the patient and their family the individualised education and care that they require [48].

Moreover, the importance of presenting information in an appropriate way is well-recognised in the literature [15]. The results of our review indicate that combining different methods of patient education can be very effective. In the review, participants highlighted the importance of obtaining information both verbally and in written form. Written teaching methods were found to be a beneficial resource for stroke patients during the chronic stroke phase [40]. If the patient identifies their favourable learning styles, professionals can educate them in a suitable manner. They can also supplement this by incorporating other types of educational materials such as online resources.

Education is paramount in ensuring effective communication. In the present study, participants reported difficulty in finding channels to contact the health services to acquire relevant information on how to manage their lives after suffering a stroke [31]. The participants highlighted dissatisfaction with healthcare providers’ communication skills in terms of clarity and effectiveness. Such communication issues can cause patients to feel uncertain and anxious. To address this, Smith et al. [49] recommend that patients and caregivers should be actively involved in the care process and should be given sufficient opportunities and follow-up to provide feedback and clarify information. Additionally, the findings indicated that long-term services, such as monitoring, assessment, and therapy, are required to improve the patients’ physical and psychosocial functioning. Martinsen et al. [31] thus suggest that follow-up services should be intensified and adapted to suit the needs of the individual patient. Moreover, as health services for stroke patients continue to be developed around the world, information and communication technology (ICT) and mobile technologies can be used to create suitable eHealth services that can ensure that patients’ care and rehabilitation needs are met [11].

Long-term care is generally considered to be more fragmented than acute stroke care [50]. Several participants have explained that, although rehabilitation services should address physical impairments, psychosocial, and social participation needs, they generally only address biomedical needs for a limited period of time [30]. A lack of healthcare resources was identified by participants as a key issue impacting the discontinuity of therapy [45]. Most participants reported that, when returning to life in the community, they found it difficult to access services, including rehabilitation services. They, thus, indicated feeling a sense of abandonment [42]. Moreover, a lack of staff was highlighted as a key factor influencing the lack of long-term care and rehabilitation options for stroke patients; the intensity of therapy programmes is directly affected by staff shortages. Moreover, the amount of time that a therapist spends with each patient ultimately determines the outcome of stroke rehabilitation. Tele-rehabilitation may be an alternative for stroke survivors who want to continue treatment and improve their quality of life [51]. Furthermore, self-management treatments, which differ from simple patient education or skills training because they are designed to motivate those with chronic conditions to take an active role in their own health management [52], can be employed. Specialised training courses for formal and informal carers are also required to enable stroke survivors and their relatives to live a healthy life [28]. In this study, stroke recovery groups in the community were identified as important and equitable resources as they give stroke survivors an opportunity to create new social networks and participate in learning, educational, and therapeutic activities [44].

### 4.1. Limitations

It is important to note that this systematic review has a number of limitations. Firstly, it is a qualitative review that generates issues such as limited generalisability. This is because most studies in the review were conducted in developed countries. Thus, it is incredibly difficult to generalise the findings to developing countries or countries with a non-Western culture. There is a lack of research that has examined the unmet needs of stroke patients in Asia which contributes to poor awareness regarding the needs of stroke patients beyond hospital settings in general, as well as poor care provisions post-discharge. Additionally, the incidence of strokes is very different between developed and developing countries, and they have different healthcare systems, policies, and challenges when it comes to stroke care provision; thus, the extent of reported unmet needs may differ substantially. Nevertheless, a wide range of stroke patients’ needs were shared across developing and developed countries, such as information needs. Further, it is also important to consider that caregivers are constantly involved in caring for the stroke patient and thus, they often provide unique perspectives. Further, strokes affect patients and caregivers differently and thus, may have different needs. Despite the importance of caregivers, their perceptions were not considered in this systematic review. Further reviews could consider integrating evidences on unmet needs from different stalk holders. In addition, the review was not able to determine how the unmet needs are affected by various levels of physical, intellectual, and personality defects. Research is needed that takes on critical stances to reveal important social, educational, ethnic, and culturally sensitive information that will improve the ability to transfer these findings to different groups of people. This is especially important since socioeconomic and cultural factors influence the outcomes and, hence, the unmet needs of stroke patients.

### 4.2. Implication for Practice

To ensure that education is sufficient and effective, healthcare workers must provide patients with relevant information on a regular basis. To achieve this, several different methods of patient education can be combined. It is also important that stroke patients are assigned a contact person who can help them to locate important information and resources, as well as answer their practical questions and concerns. Medical professionals must provide stroke survivors with discharge plans including follow-up appointments. Ideally, these follow-ups should initially take place six weeks after the discharge and then at the six-month stage. From then on, a yearly follow-up is sufficient [1]. Moreover, it is also recommended that a system for annual reviews be developed to enable changing needs to be identified and addressed. Stroke survivors and caregivers can also be given specialised training courses to promote care continuity. The data revealed in this study indicate that patients require more psychological support during the care process.

## 5. Conclusions

In this review, the needs and desires of stroke patients have been examined. The findings have indicated that the long-term needs of such patients are not being fully met at present. Moreover, the findings show that existing care plans for stroke survivors do not address the long-term needs of stroke survivors. Thus, continuous treatment and therapies are required as well as specially designed programmes that meet the long-term needs of stroke survivors. However, only three studies were included in this review, all of which were carried out in Asia and, thus, it is important to explore the topic in other continents and other settings in order to fully understand the issue at hand.

## Figures and Tables

**Figure 1 behavsci-12-00404-f001:**
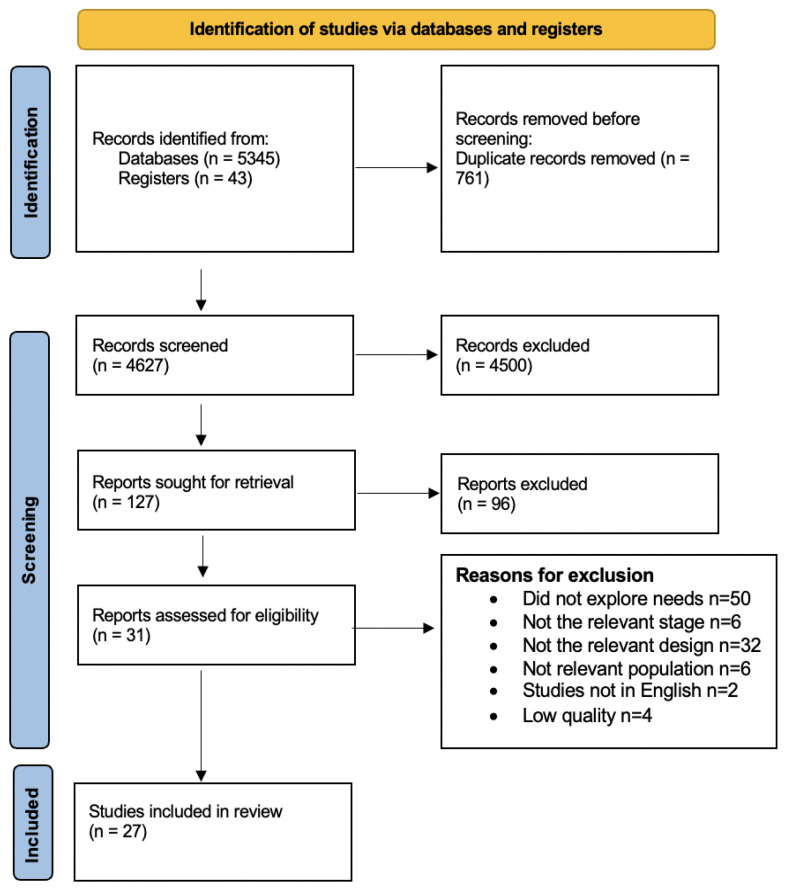
The PRISMA flow diagram.

## Data Availability

The data included in this study are available from the corresponding author upon request.

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
