# Peer review of "Exploring Stroke Patients’ Needs after Discharge from Rehabilitation Centres: Meta-Ethnography"

_behavsci, 2022, doi:10.3390/bs12100404_

Round 1
Reviewer 1 Report
This is a well-written and interesting review; however, the question of 'unmet needs' for stroke patients is open-ended and ill-defined. There are no obvious agreed guidelines, but a host of 'off the cuff' opinions, many from so-called authorities, such as the NHS and various stroke and rehab charities and other organisations, whether in the private sector or as part of the statutory arrangements for post-stroke care. In this paper the authors do not attempt to make any definition of 'post-stroke care needs', but simply report a series of qualitative opinions culled from their literature review. These are entirely unsurprising, since they reflect the subjective experiences of patients after stroke.
There is a striking absence of any attempt to consider the opinions of the immediate carers (ie family members) of patients experiencing stroke. Sadly, there is also a total absence of any consideration of the physical and intellectual'/personality defects resulting from stroke - these are potent factors in functional recovery and enjoyment of life post-stroke. For example, problems with balance, with language, with sensory perception, with released positive higher level problems such as hallucinatory behaviour and confusion are important as much as difficulties with mobility and dressing or self-care. None of these are considered as co-morbid factors.
There is no consideration of the meaning of the patients' description fo their needs - these are strikingly all-embracing, indicating the difficulty people have in understanding the nature of their deficit in function. When the brain is damaged, it is difficult for the patient to comprehend the nature of the dysfunction - in contrast to damage to a leg or an arm, for example. Damage to the 'control system' is a special category of dysfunction.
in summary, the patient description of dysfunction or dissatisfaction is entirely real, but it is not necessarily a complete description in the context of stroke.
The notion that measures to prevent future repeated stroke will be effective is unfortunately correct only in a few circumstances; eg in the context of treatable atrial fibrillation or in certain rare vasculitides. Otherwise, the results are disappointing, although few physicians would wish to directly burden a patient with such abjectly sad information.
The real missing information in this review is the 'caregiver burden' - a subject about which there are many studies. The attitude of the caregiver is crucial to the outcome in terms of happiness and adaptation.
I also miss any understanding of social status and educational level of the affected patients in these reviewed studies - this is a major factor in outcome. So much easier to recruit care for the better educated (and I am not being culturally biased in that statement, since I would add to those families with close cultural ties, for example many from India and other Asian communities - perhaps especially, in my experience, among Japanese people).
I am disturbed by the remarkable number of studies excluded form the analysis - 4500 at first glance and then many more. Surely there should be some more detailed explanation?
Finally, I find the two conclusions to be foregone ideas; there is nothing unique or new among them. And I would point out that the concept of stroke rehabilitation is open-ended, without defined endpoints, and without clear targets (the outcomes are so very variable in clinical practice). So the review adds nothing new - except a clumsy new phrase 'meta orthrography, which may be of value in the sociological literature, but which seems to be nothing very new to a neurologist used to dealing with multi-level items of information in that most complex of systems; the human mind and brain.
I suspect that the authors will be reluctant to consider most of my considerations........
Reviewer 2 Report
This manuscript presents an interesting study on stroke patients’ needs after discharge from rehabilitation centers and it is clear to see that a lot of hard work has been put into the study. While I appreciate the effort of the work presented and the significance of stroke survivors needs, I think the authors needs to improve the focus of the paper and provide more information on the results and discussion.
The paper provides very interesting data but it still needs a considerable revision to be acceptable for Behavioral Sciences Journal.
Thus, several points as indicated below need to be addressed by authors to improve the quality of the article.
In the results of this paper, four factors are extracted and classified. They are as follows:
1) Limited availability and suitability of information,
i) stroke pathology
ii) stroke recovery,
iii) information delivery
2) Adequacy of care and services
i) Motivation
a, Family support
b. Peer support
ii) Income support
iii) Travel support
3) Planning
4) Care continuity
i) Ongoing care
ii) Follow up
iii) Communication
“Limited availability and suitability of information” is related to diseases such as the pathology and prognosis of stroke. Information about pathology and prognosis of stroke should be educated during hospitalization.
Adequacy of care and services contain social problems, there is no statement concerning was established. The authors need to describe the information for these social factors from the cited studies.
Table (1)
I think it would be easier to understand if the description of the participant column and the stroke column were simplified. Clarify the representation of the time course of the “time since stroke” column. The technical terms are different in each cells.
It would be useful to include a table where the results and/or conclusions of the cited studies are listed.
Table (2)
The tables are very demanding. Table (2) would be more "reader friendly" if you exclude minute explanation.
I hope that my comment is very useful for the improvement of the article.
